# Individual and community level factors associated with modern contraceptive utilization among women in Ethiopia: Multilevel modeling analysis

**Hailay Gebrekidan**[1,2]*, **Mussie Alemayehu**[2], **Gurmesa Tura Debelew**[1]

**1** Department of Population and Family Health, Institute of Health, Faculty of Public Health, Jimma University, Jimma, Ethiopia, **2** School of Public Health, College of Health Sciences, Mekelle University, Mekelle, Ethiopia

* hailaygeb23@gmail.com

## Abstract

### Background

Modern contraceptive utilization is the most effective intervention to tackle unintended pregnancy and thereby reduce abortion and improve maternal, child, and newborn health. However, multilevel factors related to low modern contraceptive utilization and the robust analysis required for decision-making were scarce in Ethiopia.

### Objective

To investigate the individual and community-level predictors of modern contraceptive utilization among reproductive-age women in Ethiopia.

### Methods

We utilized data from a cross-sectional 2019 Performance Monitoring for Action Ethiopia survey. The survey employed a stratified two-stage cluster sampling method to select households for inclusion. In Stata version 16.0, the data underwent cleaning, aggregation, and survey weighting, following which a descriptive analysis was performed utilizing the "svy" command. Subsequently, the primary analysis was executed using R software version 4.1.3. We fitted a two-level mixed effects logistic regression model on 6,117 reproductive-age women nested within 265 enumeration areas (clusters). The fixed effect models were fitted. The measures of variation were explained by intra-cluster correlation, median odds ratio, and proportional change in variance. The shrinkage factor was calculated to estimate the effects of cluster variables using the Interval odds ratio and proportion opposed odds ratio. Finally, the independent variables with a significance level of ($P<0.05$) and their corresponding Adjusted Odds ratios and 95% confidence intervals were described for the explanatory factors in the final model.

**Data Availability Statement:** We used a cross-sectional dataset from the Performance Monitoring for Action Ethiopia (PMA-ET) 2019 survey, accessed after submitting our study's abstract to

PMA-Ethiopia at https://www.pmadata.org/countries/ethiopia. The terms and conditions of the PMA-Ethiopia data do not allow sharing the data with others (only used for the registered data). We didn't receive any special privileges in accessing the data. Thus, any researcher can apply to access the data from the PMA-Ethiopia office at https://www.pmadata.org/countries/ethiopia using the details described in this paper's Methods and Materials section.

**Funding:** The authors received no specific funding for this work.

**Competing interests:** The authors have declared that no competing interests exist.

**Abbreviations:** AIC, Akaike's Information Criterion; AOR, Adjusted Odds Ratio; CI, Confidence Interval; CPR, Contraceptive prevalence Rate; EA, Enumeration Area; EDHS, Ethiopia Demographic and Health Survey; FP, Family Planning; FRS, Female Survey Result; HHQ, Household Questionnaire; HSTP, Health Sector Transformation Plan; ICC, Intraclass Correlation Coefficient; IOR, Interval Odds Ratio; IUDs, intrauterine devices; LL, Lower Limit; MOR, Median Odds Ratio; ODK, Open Data Kit; PCV, Proportional Change in Variance; PMA, Performance Monitoring for Action; POOR, Proportion of Opposed Odds Ratios; SDG, Sustainable Development Goal; SNNP, Southern Nations, Nationalities and Peoples; TFR, Total Fertility Rate; UL, Upper Limit; VIF, Variance Inflation Factor; VPC, Variance Partition Coefficient.

## Results

In Ethiopia, the prevalence of modern contraceptive utilization was only 37.% (34.3 to 39.8). Women who attained primary, secondary, and above secondary levels of education were more likely to report modern contraceptive utilization with AOR of 1.47, 1.73, and 1.58, respectively. Divorced/widowed women were less likely to report modern contraceptive utilization (AOR:0.18, 95% CI 0.13,0.23) compared to never-married women. Discussions between women and healthcare providers at the health facility about family planning were positively associated with modern contraceptive utilization (AOR:1.84, 95% CI: 1.52, 2.23). Community-level factors have a significant influence on modern contraceptive utilization, which is attributed to 21.9% of the total variance in the odds of using modern contraceptives (ICC = 0.219). Clusters with a higher proportion of agrarian (AOR: 2.27, 95% CI 1.5, 3.44), clusters with higher literacy (AOR: 1.46, 95% CI 1.09, 1.94), clusters with empowered women and girls about FP (AOR: 1.47, 95% CI 1.11, 1.93) and clusters with high supportive attitudes and norms toward FP (AOR: 1.37, 95% CI 1.04, 1.81) had better modern contraceptive utilization than their counterparts.

## Conclusion

In Ethiopia, understanding the factors related to modern contraceptive use among women of reproductive age requires consideration of both individual and community characteristics. Hence, to enhance family planning intervention programs, it is essential to focus on the empowerment of women and girls, foster supportive attitudes towards family planning within communities, collaborate with education authorities to enhance overall community literacy, pay special attention to pastoralist communities, and ensure that reproductive-age women as a whole are targeted rather than solely focusing on married women.

## Background

Family planning allows couples and individuals to exercise their fundamental right to freely and responsibly choose the number and timing of their children [1, 2]. This practice is supported by demographic, health, and human rights considerations and plays a crucial role in empowering women, alleviating poverty, and promoting sustainable development [3]. Modern contraceptive methods have been shown to effectively prevent unintended pregnancies, leading to a decrease in maternal and child mortality rates, a reduction in unsafe abortions, and an improvement in the overall health of women and families [4, 5]. These methods are products or medical procedures that prevent reproduction from sexual intercourse [6]. It includes male and female sterilization, injectables, intrauterine devices (IUDs), contraceptive pills, implants, condoms, the standard days' method, the lactational amenorrhoea method, and emergency contraception [7].

Unintended pregnancy and births are becoming a significant public health problem that causes illness during pregnancy, complications at birth, and maternal deaths [8]. Roughly in the globe, 121 million unintended pregnancies occurred each year between 2015 and 2019, and of these unintended pregnancies, 61% (73 million) ended in abortion per year [9]. Considerably, unintended pregnancy and childbirth are higher in developing than in developed regions [8, 10]. Nevertheless, evidence suggests that implementing effective strategies such as

family planning could help prevent unintended or unwanted pregnancies in developing nations [11].

Around the world, 966 million women in the reproductive age group are currently utilizing a form of contraception, with about 874 million of them opting for modern contraceptive options. However, despite these numbers, there remain approximately 164 million women who wish to postpone or prevent pregnancy but are not practicing any form of contraception [12]. In Sub-Saharan Africa, only 56% of women who desire to avoid pregnancy utilize modern contraceptive methods, marking the region with the lowest usage rate [12]. Additionally, many of the most impoverished countries in sub-Saharan Africa continue to exhibit a substantial unmet need for family planning services [13]. Based on the findings of the 2016 Ethiopian Demographic and Health Survey (EDHS), the adoption of modern contraceptives among reproductive-age women stands at 35% for currently married individuals and 55% for sexually active unmarried women [7], figures that fall significantly short in effectively addressing the prevalent unmet needs and unintended pregnancies in Ethiopia.

To date, Ethiopia has been striving to enhance the adoption of contraception and bolster maternal, neonatal, and child health outcomes. Despite the combined endeavors of the Ethiopian government, NGOs, and the evident advantages of contraceptives, the uptake of modern contraceptive methods still remains low, with a rising unmet need reaching 22% [14] and lags behind global family planning benchmarks and objectives [15].

However, evidence has indicated that meeting all unmet needs for modern contraception in Ethiopia could lead to a 90% decrease in unintended pregnancies, subsequently resulting in a reduction of unplanned births from 1.2 million to 128,000 and a drop in abortions from 571,000 to 59,000 [16].

Sustainable Development Goal (SDG) goal 3 also emphasized universal access to sexual and reproductive health care services for healthy lives and well-being for all ages by 2030 [11]. Subsequently, Ethiopia is working towards achieving this goal and improving momentum in reaching the HSTP II and SDGs, mainly target 3.7, which focuses on ensuring access to family planning information and education by 2030 [14, 17].

Several studies elsewhere in the country revealed many reasons for low modern contraceptive use [18–21]. Most previous studies focus only on individual factors and neglect contextual factors. In addition, unmarried women, who are at a higher risk of unintended pregnancy, are frequently overlooked in research, which tends to focus on married women [22, 23], Although monitoring and reporting on modern contraceptive prevalence rates for all women have been shown to encourage the use of contraception [24].

Hence, this current study considered all reproductive-age women to represent the study population best. The primary objective of this study was to examine the individual and community factors influencing the utilization of modern contraceptives among women of reproductive age in Ethiopia, based on data from the 2019 Performance for Action Ethiopia (PMAET-2019) survey.

Moreover, previous studies' analysis only utilized the odds ratio, the proportion of change in variance (PCV), and intra-cluster correlation (ICC) to describe multilevel results. However, this current research incorporated summary measures such as Interval Odds ratio (IOR) and proportion of opposed odds ratio (POOR), as well as measures of components heterogeneity like Median odds ratio (MOR), to enhance the informativeness and explanatory power of our findings. The insights gained from this study will be valuable for policymakers, healthcare organizations, communities, and other stakeholders working towards enhancing maternal, newborn, and child health outcomes through the effective promotion of modern contraceptive methods.

## Methods and materials

### Study design, data source, and sample points

We used data from 2019 Ethiopia's National Performance Monitoring for Action (PMA) cross-sectional survey. The survey covered all regions, sample subregions and communities, and individual women [25]. For the current study, we took subset data from the national survey pertinent to the research.

The sampling of regions (the largest administrative divisions), subregions and communities (or enumeration areas (EA)), and individual women was designed centrally by Ethiopia statistical agency, ensuring representativeness at the national and sub-regional levels for both urban and rural areas. EAs were drawn with probability proportional to size (PPS) within the strata. For some of the larger regions (Tigray, Amhara, Oromia, SNNP), EAs were drawn separately for urban and rural strata within the region. For Afar, Somali, Benishangul-Gumuz (BG), Gambella, and Harari, the EAs were randomly selected with PPS in the regions, i.e., without urban and rural stratification. Addis Ababa is exclusively urban, so no stratification was used. The PMA-Ethiopia used sample weights to account for complex survey design, non-response surveys, and post-stratification for the representativeness of the samples. A total of 265 enumeration areas (clusters) of the primary sampling units were randomly selected for the PMA 2019 cross-sectional survey, and from each enumeration area, 35 households were randomly selected for the household interview.

Finally, all women of reproductive age 15 to 49 within the selected households were eligible for the female interview. Data on households and individual women of reproductive age were collected by trained enumerators who received intensive training and were overseen by supervisors. Surveys were developed on ODK, an open-source software for collecting and managing data, and loaded onto smartphones. The data were sent at the point of the interview to a central server, where they were quality-checked, allowing the enumerators to correct any errors that occurred without delay. This study's population was all women's reproductive age group (15–49 years). The details of the PMA-Ethiopia survey design and sampling techniques are described elsewhere [26].

### Sample

First, 9107 women of reproductive age 15–49 years were identified from the PMA-ET 2019 survey to identify the eligible study population for this study. Then, we dropped incomplete household and female interviews. The following Stata command criteria were used to identify the female respondents for this study. Firstly, keep if the woman is part of a household that completed the household survey (HHQ_result = = 1) and completed the female survey (FRS_result = = 1). So, incomplete household and female questionnaire interviews were deleted using the Stata command keep if HHQ_result = = 1 & FRS_result = = 1 [27]. Accordingly, 8928 remained. Then, pregnant women were excluded as they were not eligible for modern contraceptive method use. Sexually active women were also selected by excluding those who never had sexual intercourse. Finally, after excluding the menopause and infecund women, a total of 6117 eligible women were identified and included in the analysis [28]. Details are indicated in (Fig 1).

### Measures of variables

The outcome variable in this study was modern contraceptive users/non-users. Based on the previous related literature, around 14 (fourteen) independent variables were selected for the analysis [29–31]. In addition, we created six community-level variables by aggregating some of

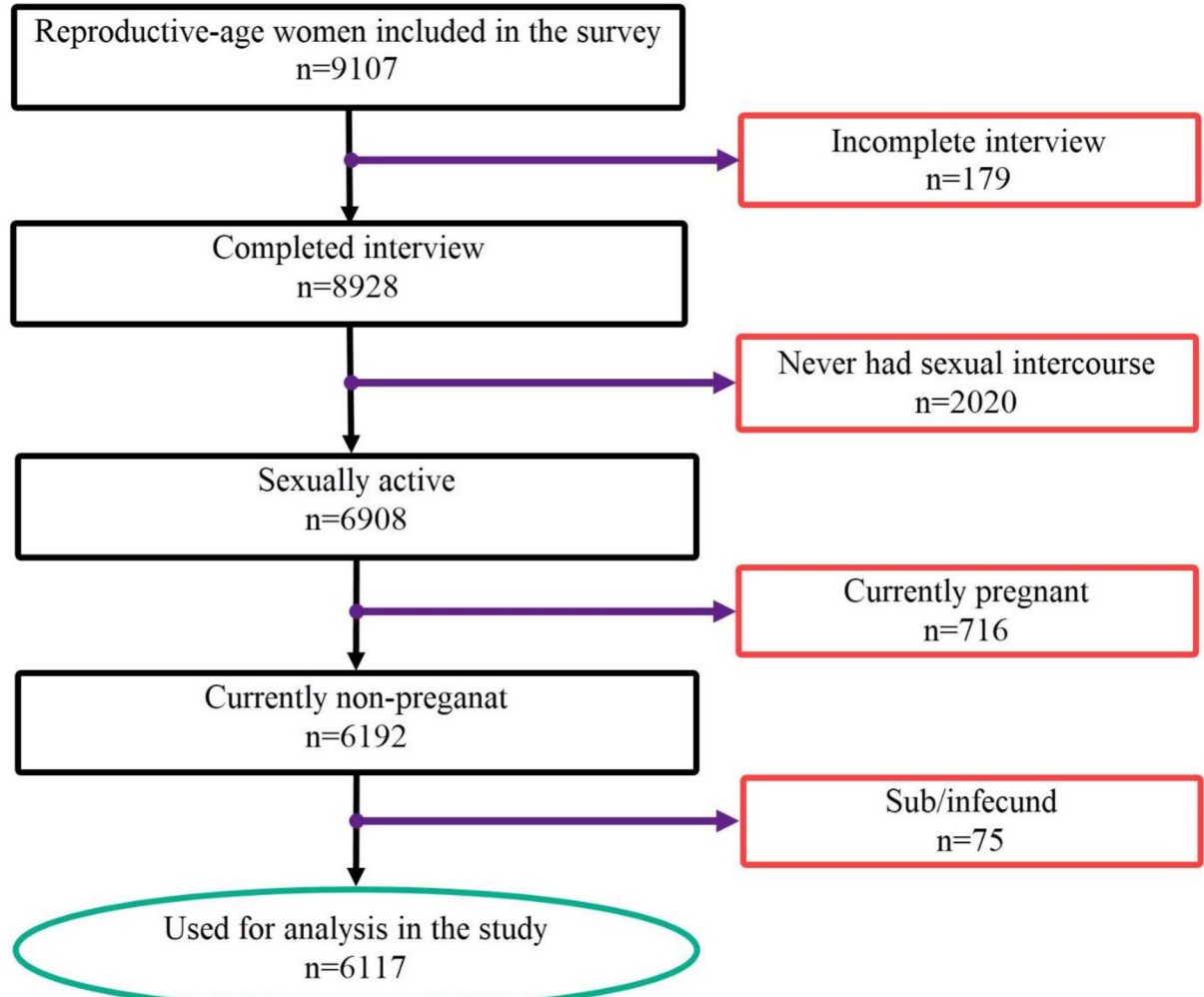

**Fig 1. Selection of eligible women for modern contraceptive utilization among reproductive-age women in Ethiopia, data from PMA-ET, 2019.**

the individual-level variables. The woman was considered a "user" if she had been utilizing any modern methods of contraceptives such as female sterilization, intrauterine contraceptive devices, injectables, implants, pills, male and female condoms, lactational amenorrhea method, standard days method, emergency contraception during the 2019 PMA-ET survey period while a woman who had been utilizing traditional or no method was considered as a "non-user".

The independent variables were categorized as level I (individual women variables) and level II (cluster/community level variables).

### Individual-level variables

The level 1 (individual women level) variables included age, educational status, marital status, marriage history, place of birth, parity, Educational Status, religion, the intention to have the next birth, household wealth quintile, knowledge of contraceptive methods, whether visited a health facility in the past 12 months and discussed family planning, whether visited by a health worker in the past 12 months (Table 1).

**Table 1. Descriptions of the dependent and independent variables considered in this study.**

| Variables | Descriptions | Measurements |
|---|---|---|
| Dependent variable | | |
| Modern contraceptive utilization | The modern contraceptive utilization among reproductive-age women at the survey time | A woman who was using any one of the modern contraceptive methods during the survey time was coded as "1," and who was not used as "0." |
| Independent level variables | | |
| Age | Age of women in complete years at the interview time | Categorized for analysis into 15–19, 20–34, and 35–49 years. |
| Educational Status | The highest level of education a woman has attained | Categorized into four groups: "No education," "Primary," "secondary," and "More than secondary," |
| Marital status | Marital status of women at the interview time | It is categorized into "Never married," "Married or in union/ relationship " and "Divorced/widowed. |
| Visited by the health worker | A woman was visited by HEW who talked about FP in the past 12 months | A woman who was visited by HEW and talked about FP in the past 12 months was coded as "1" and not as "0." |
| Wealth quintiles | Household assets, home, land, and livestock ownership were assessed, and the wealth index was computed by using principal component analysis (PCA) | The wealth status was categorized into five groups: "Lowest quantile," "Lower quantile," "Middle quantile," "Higher quantile," and "Highest quantile." |
| Community level Variables | | |
| Women and girls empowerment about FP | The women were asked to respond to five empowerment-related questions. | Accordingly, as a result, it is normally distributed; the mean score was computed for the women and girls' empowerment about FP, and it was categorized below the mean as "Not Empowered" and mean and above the mean as "Empowered." |
| Exposure to family planning information | The respondents were asked five exposures to family planning information-related questions. E.g. In the last few months, have you heard about family planning on the radio?? The response was "yes" or "no". Then, its answer is summed up to form one continuous variable. | Accordingly, as a result, the mean score was normally distributed. The mean score was computed for exposure to family planning information, and it was categorized below the mean as "Lower exposure" and mean and above the mean as "higher exposure." |
| Community way of life | The community's way of life was categorized as pastoralist based on the Ethiopian region category. Based on that, regions 1, 3, 4, 7, 9, 10, and 11 were categorized as agrarian, and regions 2, 5, 6, and 8 were classified as pastoralist communities. | Hence, the communities were categorized into "Pastoralist" and "agrarian." |
| Community attitude/norms towards contraceptives | The women were asked to respond to five community attitudes/ norms towards family planning questions. Some of the items were "Women who use family planning are considered promiscuous." The response was scored as "strongly agree," "somewhat agree," "Neither agree nor disagree," "somewhat disagree," and "strongly disagree." Items negatively worded were reversed and scored. Then, the response is summed up to generate a continuous variable. | Accordingly, as a result, it was normally distributed, and the mean score was computed for Community norms about FP. It was categorized below the mean as "Non-supportive norm" and above the mean as "Supportive norm." |
| Community knowledge of the modern contraceptive method | The women were asked whether they had heard of each of the 11 modern methods of contraceptives, and a composite knowledge score was produced using PCA. The response to 11 items was labeled "yes" and "no." Examples of items include: "Ever heard of female sterilization." | We used the median score as a cutoff point to categorize the composite knowledge. The result was not normally distributed. So, the response was classified as having "Good knowledge of modern contraceptives" for those who scored the median and above the median and "Poor knowledge of modern contraceptives" for those who scored below the median. |
| Community literacy level | It was measured by asking the respondents about the highest school they attended and categorized into "lower literacy" and 'higher literacy." | Accordingly, as a result, the mean score was normally distributed for the community literacy level, and it was categorized below the mean as "lower literacy" and mean and above the mean as "higher literacy." |

## Community-level variables

Community-level variables are created by aggregating the individual-level variables. Based on the mean/median value of the response, the variable was labeled as "lower" and "higher" categories for the cluster that scored below mean, mean, and above mean value, respectively (Table 1).

## Statistical analysis

The data obtained from the official PMA database was exported to Stata version 16.0. Subsequently, Stata version 16.0 was utilized for tasks such as data cleaning, aggregating community-level variables from individual-level variables, and carrying out descriptive analysis using the "svy" command. This approach addressed clustering, ensured representativeness, and obtained accurate estimates. Afterward, descriptive statistics such as frequency distributions and percentages were used to analyze a range of individual and community-level variables. Then, the primary analysis was carried out using R software version 4.1.3.

Since the reproductive-age women were nested within the Enumeration areas/cluster, we fitted a two-level mixed effects logistic regression model on 6,117 women nested within 265 enumeration areas (clusters). Four models were fitted: empty Model I, Model II, Model III, and Model IV. Multivariable two-level logistic regression analysis was performed, and the fixed effect models were fitted and described as adjusted odds ratios with 95% confidence intervals. The extent of variation was explained by intra-cluster correlation, median odds ratio, and proportional change in variance. The shrinkage factor was calculated to estimate the Interval odds ratio and proportion opposed odds ratio. The deviance and Akaike information criteria were used as model fitness criteria.

**Model I (Empty model).**    We run the model without explanatory variables. This aimed to test the random effect of the between-cluster variability. The intra-class correlation (ICC) was estimated to explain if the data justified using a multilevel approach for analyses by depicting the level of variability between clusters.

**Model II.**    Examined the effects of individual-level variables on modern contraceptive utilization. The variables that showed statistical significance at the individual level in the bivariate multilevel logistic regression analysis were incorporated into this model.

**Model III.**    Examined the effect of community-level variables on modern contraceptive utilization. Those community-level variables, which were statistically significant in the bivariate multilevel logistic regression analysis, were included in this model. They were formed by aggregating the variables from the individual-level variables.

**Model IV.**    The fourth step is model building for the multiple multilevel logistic regression analysis. The statistically significant variables in models II and III were included in this model. This final model identified average associations between specific individual characteristics and community-level variables at the same time while adjusting for clustering [32].

The multilevel analysis of individual heterogeneity approach combines both the multilevel analysis of associations for the estimation of specific contextual effects and the multilevel analysis of variance (e.g., the degree of clustering, ICC) for the investigation of general contextual effects (i.e., non-specific contextual influences on modern contraceptive user/non-user [33, 34].

For cluster-level variables, the concept of an intra-cluster association is not as straightforward as in the case of individual-level variables because the value of cluster-level variables is constant for all individuals in the cluster. Therefore, one needs to consider assuming the existence of several clusters. In addition, various alternatives to the ICC for binary responses have been proposed to quantify the extent of general contextual effects. These include the Shrinkage factor, IOR (Interval odds ratio), POOR (proportional opposed odds ratio), MOR (Median Odds ratio), and PCV (variance partition coefficient) [35].

The shrinkage factor was calculated to adjust the marginal or population average regression coefficients and odds ratios for the higher-level unit variability, considering the conditional or cluster-specific interpretation of the regression coefficients. So, it shrinks the estimated coefficients toward the overall average, reducing the impact of extreme or spurious estimates. A shrinkage factor close to 1 indicates minimal shrinkage, while values closer to 0 imply

substantial shrinkage. It is used to calculate interval odds ratio (IOR) and proportion of opposed odds ratio (POOR), which are summary measures of the effects of cluster-level variables [23, 24].

The IOR provides a range within which the true odds ratio of an individual unit from a given level is likely to fall. It is a fixed-effects measure for quantification of the effect of higher-level units on the outcome of interest. It is defined as an interval for odds ratios covering the middle 80 percent of the odds ratios of different cluster-level covariates or the middle 80% range of the distribution of odds ratios formed by randomly comparing clusters exposed and non-exposed to the contextual variable. The interval is narrow if the between-cluster variation is small and wide if the between-cluster variation is large. If the interval contains 1, the cluster variability is large compared to the effect of the cluster-level variable. If the interval does not contain 1, the effect of the cluster-level variable is large compared to the unexplained between-cluster variation [36, 37].

An alternative to the IOR-80% is the Proportion of Opposed Odds Ratios (POOR). That is the proportion of odds ratios in the opposite direction to the overall OR. The values of the POOR extend between 0% and 50%. A POOR of 0% means all odds ratios have the same sign. A POOR of 50% means that half of the odds ratios are of the opposite sign, so the association is very heterogeneous. A higher POOR indicates a stronger opposition to the outcome, while a lower POOR suggests a weaker opposition [37].

The estimated coefficients and odds ratios of the IOR and POOR above do not provide a measure of general contextual effects. So, we need to measure the components of variance (the variance partition coefficient (VPC) or intra-class correlation (ICC)) and the median odds ratio (MOR) to determine the variation or heterogeneity in outcomes between clusters. The VPC/ICC measures the proportion of total variance due to the difference between groups or between-cluster variation. The higher this proportion, the higher the general contextual effect. The VPC/ICC is a crucial concept in the multilevel logistic regression analysis as it quantifies the proportion of observed variation in the outcome attributable to the clustering effect [38].

The VPC (ICC) measures components of variance (clustering) that consider both between- and within-cluster variance. The VPC value ranges from zero to one. If all subjects in the same cluster exhibited the same response, it has a value of 1, and it would be 0 if there were no within-cluster homogeneity of responses [39]. In line with this, a proportional change in cluster variation (PCV) was calculated for all models by taking the empty model as a reference point. Percentage Change in Variance (PCV) indicates the proportion of the total observed individual variation of modern contraceptive utilization attributable to cluster variations. The MOR is a measure of heterogeneity mainly dependent on between-cluster variance. It revealed the median value of the odds ratios for all possible pairs of individuals from different clusters and was used to assess the degree of clustering or heterogeneity in modern contraceptive utilization [40]. The higher MOR indicates greater variability in odds ratios between clusters, suggesting that individual-level factors may not fully explain the outcome variation and that important contextual or group-level factors may be at play. Conversely, the lower MOR suggests less variability between clusters and a stronger influence of individual-level factors on the outcome [35]. The measure is always greater than or equal to 1. If the MOR is 1, there is no variation between clusters (no second-level variation). The MOR will be large if there is considerable between-cluster variation [36].

Multi-Co linearity among the independent variables was checked using the tolerance and the Variance Inflation Factor (VIF). Model comparison was made using Akaike's Information Criterion (AIC) tests. The model with the lowest value of AIC was considered a better-fit model. Adjusted odds ratios and associated 95% confidence intervals (CIs) were presented for all three models except model I. Finally, the independent variables with a significance level of

(P < 0.05) and their corresponding Odds ratios and 95% confidence intervals were described for the explanatory factors in the final model.

## Ethical considerations

This study did not require ethical approval since we have used publicly available PMA-Ethiopia datasets. The data were accessed from http//www.pmadata.org/countries/ethiopia after the purpose of the study with respect to the data-sharing policy was explained. The accessed data were used for the purpose of a registered research paper only. Confidentiality of the data was maintained, and no effort was made to identify any household or individual respondents interviewed in the survey. The data were not passed on to other researchers without the written consent of PMA-Ethiopia.

## Results

### Socio-demographic and reproductive (Individual-level) characteristics of the respondents

This study focused on a sample of 265 clusters (Enumeration areas)in Ethiopia. We included unweighted 6117 or weighted 6065 reproductive-age women in the 265 clusters (Enumeration areas). The prevalence of modern contraceptive method use among the respondents was 37% (34.3–39.8). However, more than 97% of the respondents have heard of at least one modern contraceptive method. About 1588 (43.4%) of non-current users of contraceptive methods have a future intention to use modern contraceptive methods. Around 2,812.9 (46.4.%) of the respondents have no education. Of these, 2480(96%) could not read and write a simple sentence. More than half, 3343(55.13%) of the respondents were aged 20–34. About 5007(82.56%) respondents were married/living with a partner. About 1403.6 (23.14%) respondents had six or more children. The proportion of respondents who preferred wait duration from now before another childbirth was soon/now was 2711(68.15%) (Table 2).

### Community-level characteristics (higher-level variables) of the respondents

Two hundred sixty-five clusters (Enumeration areas) were considered a unit of analysis. Accordingly, 52% and 59.6% of the clusters had lower community literacy levels and exposure to FP information, respectively. Over half (52.8%) of the clusters have empowered women and girls about FP. Half (49.5%) of the clusters have a non-supportive attitude/norm toward family planning. More than half (54.2%) of the clusters had poor knowledge of modern contraceptive methods, and 15.2% of the clusters' community way of life belonged to pastoralist communities (Table 3).

### Multivariable multilevel logistic regression analyses

**Model building and selection.** Of the four models (I-IV) that we fitted, model IV best fits the data. The random effect results were estimated using intra-class correlation (ICC), variance partition coefficient (VPC), a proportional change in variance (PCV), shrinkage factor, Interval odds ratio (IOR), and proportion opposed odds ratio (POOR), Median odds ratio (MOR). Thus, a two-level mixed-effects logistic regression model was considered for further analysis to consider all the above.

As shown in the table below for the empty model, 21.9% of the total variance in the odds of using modern contraceptives was attributed to the between-cluster variation (ICC = 0.219). The result of the random effect model showed that the intra-class correlation coefficient (ICC) was 21.9%, indicating that the correlation between women in the same EA on modern

**Table 2. Socio-demographic and reproductive (individual-level) characteristics of respondents, data from PMA-Ethiopia, 2019 (n = 6117).**

| | Unweighted | | | | Weighted | | | |
|---|---|---|---|---|---|---|---|---|
| Individual level Characteristics | N | % | 95% CI | | N | % | 95%CI | |
| | | | LL | UL | | | LL | UL |
| Educational status of women | | | | | | | | |
| No education | 2600 | 42.5 | 41.27 | 43.75 | 2812.9 | 46.4 | 43.7 | 49.08 |
| Primary | 2034 | 33.25 | 32.08 | 34.4 | 2074 | 34.2 | 31.9 | 36.56 |
| Secondary | 883 | 14.4 | 13.57 | 15.3 | 715.6 | 11.8 | 10.5 | 13.17 |
| More than secondary | 600 | 9.8 | 9.08 | 10.57 | 462.1 | 7.62 | 6.3 | 9.17 |
| Literacy for those with no education (n = 2600) | | | | | | | | |
| No | 2489 | 95.7 | 94.8 | 96.4 | 2480 | 96 | 94.3 | 97.35 |
| Yes | 111 | 4.3 | 3.5 | 5.1 | 100 | 4 | 2.6 | 5.6 |
| Residence | | | | | | | | |
| Urban | 2537 | 41.47 | 40.2 | 42.7 | 706 | 11.63 | 29.1 | 34.13 |
| Rural | 3580 | 58.53 | 57.28 | 59.7 | 5360 | 88.37 | 65.87 | 70.8 |
| Religion | | | | | | | | |
| Orthodox Christian | 3008 | 49.1 | 47.9 | 50.4 | 2880 | 47.47 | 42.88 | 52.1 |
| Muslim | 1747 | 28.5 | 27.4 | 29.7 | 1752 | 28.89 | 23.62 | 34.79 |
| Protestant | 1230 | 20.1 | 19.1 | 21.1 | 1335.7 | 22.02 | 18.45 | 26.06 |
| Other | 132 | 2.1 | 1.8 | 2.5 | 98 | 1.62 | 0.9 | 2.78 |
| Age | | | | | | | | |
| 15–19 | 444 | 7.25 | 6.6 | 7.93 | 456.7 | 7.53 | 6.5 | 8.69 |
| 20–34 | 3487 | 57 | 55.7 | 58.2 | 3343 | 55.13 | 53.16 | 57.07 |
| 35–49 | 2186 | 35.74 | 34.5 | 36.9 | 2265 | 37.34 | 35.4 | 39.3 |
| Marital Status | | | | | | | | |
| Never married | 363 | 5.9 | 5.37 | 6.55 | 277.7 | 4.58 | 3.81 | 5.49 |
| Married or in a union/relationship | 4885 | 79.8 | 78.8 | 80.8 | 5007 | 82.56 | 81.02 | 83.99 |
| Divorced/widowed/widower | 869 | 14.2 | 13.3 | 15.1 | 780 | 12.86 | 11.66 | 14.17 |
| Wealth Index quantile | | | | | | | | |
| Lowest quintile | 1002 | 16.38 | 15.47 | 17.3 | 1163.7 | 19.19 | 16.05 | 22.8 |
| Lower quintile | 1027 | 16.79 | 15.87 | 17.7 | 1204 | 19.85 | 17.9 | 21.9 |
| Middle quintile | 1027 | 16.79 | 15.87 | 17.7 | 1191 | 19.64 | 17.28 | 22.2 |
| Higher quintile | 1239 | 20.25 | 19.26 | 21.28 | 1228.7 | 20.26 | 17.1 | 23.8 |
| Highest quintile | 1821 | 29.77 | 28.64 | 30.9 | 1227 | 21.06 | 18.6 | 23.7 |
| Total live birth | | | | | | | | |
| 0 | 887 | 14.5 | 13.6 | 14.4 | 752.3 | 12.4 | 11.28 | 13.6 |
| 1 | 1184 | 19.4 | 18.3 | 20.4 | 1121.5 | 18.49 | 17.25 | 19.79 |
| 2–3 | 1739 | 28.4 | 27.3 | 29.6 | 1614.8 | 26.62 | 24.9 | 28.38 |
| 4–5 | 1128 | 18.4 | 17.4 | 19.4 | 1172.8 | 19.34 | 18 | 20.71 |
| 6–14 | 1179 | 19.3 | 18.3 | 20.3 | 1403.6 | 23.14 | 21.2 | 25.17 |
| age at first sex | | | | | | | | |
| 9–15 | 2183 | 35.68 | 34.49 | 36.9 | 2318.8 | 38.23 | 35.36 | 41.18 |
| 16–24 | 3550 | 58 | 56.8 | 59.2 | 3454.5 | 56.96 | 54.1 | 59.7 |
| 25–39 | 384 | 6.27 | 5.7 | 6.9 | 291.7 | 4.81 | 4.19 | 5.5 |
| Ever been pregnant | | | | | | | | |
| No | 814 | 13.3 | 12.48 | 14.18 | 684.4 | 11.28 | 10.2 | 12.4 |
| Yes | 5303 | 86.7 | 85.8 | 87.5 | 5380.6 | 88.72 | 87.5 | 89.77 |
| Preferred wait duration from now before another childbirth (n = 4,011) | | | | | | | | |

*(Continued)*

**Table 2.** (Continued)

| Individual level Characteristics | Unweighted | | | | Weighted | | | |
|---|---|---|---|---|---|---|---|---|
| | N | % | 95% CI | | N | % | 95%CI | |
| | | | LL | UL | | | LL | UL |
| Don't know | 283 | 7.1 | 6.3 | 7.9 | 1057 | 26.57 | 24.57 | 28.67 |
| Soon/now | 2616 | 65.2 | 63.7 | 66.7 | 2711 | 68.15 | 65.9 | 70.3 |
| Years | 1112 | 27.7 | 26.3 | 29.1 | 209.9 | 5.28 | 4.2 | 6.5 |
| Women visited by HEW, who talked about FP in the past 12 months | | | | | | | | |
| No | 5362 | 87.7 | 86.8 | 88.4 | 5387.17 | 88.82 | 87 | 90.4 |
| Yes | 755 | 12.3 | 11.5 | 13.2 | 677.8 | 11.18 | 9.6 | 12.9 |
| Attended a group FP counseling session with a provider in the past 12 months | | | | | | | | |
| No | 5459 | 89.2 | 88.4 | 90 | 5433 | 89.58 | 87.3 | 91.5 |
| Yes | 658 | 10.8 | 10 | 11.5 | 631.9 | 10.42 | 8.5 | 12.68 |
| Staff spoke about FP methods at the facility visit (n = 3761) | | | | | | | | |
| No | 2,872 | 76.4 | 75 | 77.7 | 2830.8 | 75.81 | 72.69 | 78.67 |
| Yes | 889 | 23.6 | 22.3 | 25 | 903.2 | 24.2 | 21.3 | 27.3 |
| Will you use family planning in the future (not current users) (n = 3687) | | | | | | | | |
| No | 2139 | 58 | 56.4 | 59.6 | 2067.3 | 56.56 | 53.8 | 59.26 |
| Yes | 1548 | 42 | 40.4 | 43.6 | 1587.7 | 43.4 | 40.7 | 46.17 |
| Prevalence of modern contraceptive utilization | | | | | | | | |
| Non-user | 4007 | 65.5 | 64.3 | 66.7 | 3818.7 | 62.96 | 60.16 | 65.68 |
| User | 2110 | 34.5 | 33.3 | 35.7 | 2246 | 37.04 | 34.3 | 39.8 |

contraceptive use was 0.219, i.e., 21.9% of the variation in the utilization of modern contraceptives among reproductive age women could be attributed to EA difference.

In the final model, the proportional change of the variance (PCV) of the combined factors was 20.5% ($\tau$ = 0.732). This indicated that individual and community-level predictors simultaneously explained 20.5% of the variation in utilizing modern contraceptives across communities. Likewise, a shrinkage factor was calculated better to interpret the intra-cluster association in the cluster level factor. Model III's shrinkage factor value was 0.893, which is nearly one. This indicates that the population-average regression coefficients (and corresponding population-average odds ratios) for the four clusters/community-level factors are very close to the conditional or cluster-specific regression coefficients and corresponding odds ratios. The 80% Interval odds ratio (IOR) for the four community-level factors in model III were (0.48, 10.7) for pastoralist vs. agrarian community way of life and (0. 308, 6.87) for lower vs. higher community literacy level. Besides, it was (0. 31, 6.927) for un-empowered vs. empowered women and girls about FP, (0.29, 6.46) for community non-supportive attitude/norm vs. supportive attitude/norm towards FP. Overall, as the interval contains 1, the cluster variability is large compared to the effect of the cluster-level variable.

In addition, the proportion opposed odds ratio (POOR) of the community factors value was 0.248 for the community way of life, 0.378 for the community literacy level, 0.375 for women and girls' empowerment about FP, and 0.397 for community attitude/norms towards FP. Thus, in 37.5% of comparisons between the community with non-empowered and empowered women and girls about FP, the OR for this comparison would be different in

**Table 3. Community-level variables of respondents, data from PMA-Ethiopia, 2019 (n = 6117).**

| Community level Characteristics | No women (%) | 95% CI | | No of clusters |
|---|---|---|---|---|
| | | LL | UL | |
| Community literacy level | | | | |
| lower | 3181 (52) | 50.7 | 53.2 | 138 |
| higher | 2936 (48) | 46.7 | 49.2 | 127 |
| Community way of life | | | | |
| Pastoralist | 927 (15.2) | 14.2 | 16 | 40 |
| Agrarian | 5190 (84.8) | 83.9 | 85.7 | 225 |
| Exposure to FP information | | | | |
| Low exposure | 3644 (59.6) | 58.3 | 60.8 | 158 |
| High exposure | 2473 (40.4) | 39.2 | 41.6 | 107 |
| Women and girl's empowerment about FP | | | | |
| Not empowered | 2889 (47.2) | 46 | 48.4 | 125 |
| empowered | 3228 (52.8) | 51.5 | 54 | 140 |
| Community attitude/norms towards FP | | | | |
| Non-supportive attitude/norm | 3029 (49.5) | 48.2 | 50.7 | 131 |
| Supportive attitude/norm | 3088 (50.5) | 49.2 | 51.7 | 134 |
| Community knowledge of the modern contraceptive method | | | | |
| Poor | 3317 (54.2) | 52.9 | 55.4 | 144 |
| Good | 2800 (45.8) | 44.5 | 47 | 121 |

direction from that of the overall OR at the community level. Moreover, in a 37.5% pairwise comparison, the odds of modern contraceptive utilization would be higher in the community with empowered women and girls about FP than in non-empowered. Further, in the model, the Median Odds ratio (MOR) was 2.5 (empty model), 1.65 (individual factors), 2.19 (community factors), and 2.26 (combined model). The MOR for individual factors was 1.65, with a reciprocal of 0.6. Three out of four individual-level factors had odds ratios falling within the interval of 0.6 to 1.65, indicating consistent effects across different clusters. These factors did not significantly deviate from one, suggesting a limited impact on variation in modern contraceptive utilization between clusters. Therefore, individual-level factors do not strongly influence differences across clusters, and cluster effects play a more dominant role in explaining variability. Additionally, clustering notably affected the final model, as the MOR for unexplained community variation decreased from 2.5 in the null model to 2.26 in the final model when all predictors were considered (Table 4).

## Individual and community-level determinants of modern contraceptive utilization

**Effect of individual-level characteristics on modern contraceptive utilizations.** After adjusting for the individual and community factors, reproductive-age women with the marital status of divorced/widowed were less likely to report modern contraceptive use (AOR:0.18, 95% CI 0.13,0.23) than never-married women. The data also showed that the odds ratio of modern contraceptive utilization increases with an increasing level of education. For instance, women who attended primary school, secondary school, and more than secondary were more likely to report modern contraceptive utilization (AOR: 1.47, 95% CI 1.20, 1.79), (AOR: 1.73, 95% CI 1.33, 2.26) and (AOR: 1.58, 95% CI 1.16, 2.17), respectively than with no education. Moreover, women who reported having facility visits for family planning discussions in the

**Table 4. Measures of the effects and heterogeneity of community-level variables (Variance, a proportional change in variance, Shrinkage factor, interval odds ratio, proportion of opposed odds ratio, and the median odds ratio), data from PMA-Ethiopia, 2019 (n = 6117).**

| Variable | Empty model | Individual Factors | | Community Factors | | | Combined Factors |
|---|---|---|---|---|---|---|---|
| | | IOR lower | IOR Upper | IOR lower | IOR Upper | POOR | |
| Community-level factors | | | | | | | |
| Community way of life | | | | 0.48 | 10.7 | 0.248 | |
| Community literacy level | | | | 0.308 | 6.87 | 0.378 | |
| women and girls' empowerment about FP | | | | 0.31 | 6.927 | 0.375 | |
| Community attitude/norms towards FP | | | | 0.29 | 6.46 | 0.397 | |
| Variance ($\tau^2$) | 0.928 | 0.278 | | 0.679 | | | 0.732 |
| Shrinkage factor | 0.893 | 0.955 | | 0.899 | | | 0.893 |
| VPC/ICC | 0.219 | 0.077 | | 0.171 | | | 0.182 |
| MOR | 2.5 | 1.653 | | 2.194 | | | 2.262 |
| PCV | Ref | 69.8% | | 26.4% | | | 20.5% |
| AIC | 7391.5 | 3366.9 | | 7319.0 | | | 4295.42 |

AIC: Akaike Information Criterion, VPC: variance partition coefficient, ICC: Intra-class correlation, PCV: proportional change of the variance, IOR: Interval odds ratio, POOR: the proportion of odds ratios in the opposite direction, MOR: median odds ratio

last 12 months were more likely to report modern contraceptive use (AOR:1.84, 95% CI 1.52, 2.23) compared to never having had a visit in the last 12 months.

**Effect of community-level factors on modern contraceptive utilizations.** Accordingly, after controlling the effect of individual and community factors, clusters with a high number of agrarian community ways of life were reported to have better modern contraceptive utilization (AOR: 2.27, 95% CI 1.5, 3.44) and higher community literacy levels were reported to have better modern contraceptive utilization (AOR: 1.46, 95% CI 1.09, 1.94). Similarly, clusters that empowered women and girls about FP were more likely to report FP use (AOR: 1.47, 95% CI 1.11, 1.93) than non-empowered clusters. Besides, clusters with supportive community attitudes/norms toward FP were associated with higher odds of modern contraceptive use (AOR: 1.37, 95% CI 1.04, 1.81) than clusters with non-supportive community attitudes/norms towards FP (Table 5).

## Discussion

This research indicated that the use of modern contraceptives among women of reproductive age was low, with only 37% utilizing such methods. This figure is consistent with the findings of the EDHS 2016 report [7]. However, it is higher than the rates reported in Sub-Saharan Africa's demographic and health surveys [41]. Despite this, Ethiopia has not met the ambitious targets set by FP2020 to increase the contraceptive prevalence rate to 55% and the total fertility rate to three [15]. To achieve SDG target 3.7, significant efforts are required to improve access to and utilization of modern contraceptive methods in the country, as there is still a long way to go in meeting its family planning objectives.

The current study found that modern contraceptive use is influenced by individual factors such as marital status, education level, and discussions about family planning at health facilities. Community factors such as supportive norms, empowerment of women and girls in family planning, agrarian lifestyle, and higher literacy levels were also associated with increased modern contraceptive use. Together, these individual and community factors accounted for around 20.5% of modern contraceptive utilization among women of reproductive age in Ethiopia.

**Table 5. Multilevel mixed-effects logistic regression modeling of individual and community level factors associated with modern contraceptive utilization among reproductive-age women in Ethiopia, data from PMA-Ethiopia 2019 (n = 6117).**

| Variable | Odds ratio (95% CI) | | | |
|---|---|---|---|---|
| | Empty model | Individual-level factors | Community-level factors | Combined factors |
| Individual-level factors | | | | |
| Birth events | | 0.91(0.82,1.01) | | |
| Marital status: Divorced/widowed | | 0.2(0.15,0.27) *** | | 0.18(0.13,0.23) *** |
| School Attended: Primary | | 1.25(1,1.56) * | | 1.47(1.20,1.79) *** |
| School Attended: Secondary | | 1. 36(1.01,1.81) * | | 1.73(1.33,2.26) *** |
| School Attended: More than secondary. | | 1.03 (0.75,1.42) | | 1.58(1.16,2.17) ** |
| Marriage history: More than once | | 0.73 (0.58,0.92) ** | | 0.89(0.71,1.11) |
| Visited by a health worker | | 1.25(0.97,1.61) | | |
| Visited group counseling | | 1.89(0.67,1.17) | | |
| Facility FP discussion | | 1.6(1.29,1.97) | | 1.84(1.52,2.23) *** |
| Community-level factors | | | | |
| Community knowledge of the modern contraceptive method | | | 1.02(0.76,1.99) | |
| Community attitude/norms towards FP | | | 1.39((1.09,1.78) ** | 1.37(1.04,1.81) * |
| Women and girl's empowerment about FP | | | 1.53(1.2,1.95) *** | 1.47(1.11,1.93) ** |
| Exposure to family planning information | | | 0.96(0.7,1. 32) | |
| Community way of life | | | 2.55(1.77, 3.68) *** | 2.27(1.5, 3.44) *** |
| Community literacy level | | | 1.62(1.22,2.16) *** | 1.46(1.09,1.94) ** |

Significant at p-value 0 '***' 0.001 '**' 0.01 '*' 0.05

In this research, it was observed that women who are divorced or widowed were less likely to use modern contraceptives in comparison to those who have never been married. Studies conducted in sub-Saharan Africa highlighted a higher prevalence of contraceptive use among never-married women than divorced/widowed or married women [42]. Additionally, research indicated that married adolescent girls exhibit the lowest median prevalence of modern contraceptive use across various global regions [43]. This trend could be attributed to the fact that divorced/widowed women and married adolescents may not have reached their desired family size, while the infrequent sexual activity among divorced/widowed women might lead to a lower likelihood of contraceptive utilization. This collective data suggests that being married does not necessarily pledge to higher contraceptive utilization among women. As a result, family planning initiatives need to target reproductive-age women comprehensively rather than solely focusing on married women.

We found that the odds ratio of modern contraceptive utilization rises with higher levels of education, peaking at secondary education. However, there was a slight decrease in contraceptive use among women with education beyond the secondary level compared to those with secondary education. A study done in Nigeria revealed that increasing levels of education rise modern contraceptive utilization [44]. Many studies also show that a woman's educational attainment is positively associated with contraception use [45–47]. Moreover, education alone was indicated to have had a significant effect on the utilization of contraception [48]. Women were less likely to utilize modern contraceptives without education [41]. The possible reason for the decrement in modern contraceptive utilization in those above secondary education might be due to late marriage after the completion of higher education and the desire for more children at later ages.

Having family planning discussions at a health facility was positively associated with modern contraceptive utilization. This aligns with a study done in Uganda, which discussed FP

with a field or health worker and remained statistically significant [49]. Women were less likely to utilize modern contraceptives if they were not told of family planning at a health facility [41]. Hence, as women have more contact with healthcare providers, their chances of receiving information and using contraceptives will increase.

Thus, the women who get an excellent opportunity to be aware of FP would be vital to fill the missed opportunity [7]. In addition, it creates a conducive environment that helps mitigate the myths and misconceptions about FP and enhances the relationship between the health care providers and women, thereby increasing FP utilization. Such collective effort would contribute to increasing women's actual modern contraceptive utilizationCommunity norms/attitudes towards FP utilization were a community-level factor influencing contraceptive utilization. Our finding aligns with another study done in Ethiopia and 73 low-and middle-income countries [33, 40]. In this study, women from clusters with supportive attitudes/norms toward family planning were associated with modern contraceptive utilization. Moreover, another study revealed that social pressure on women to bear and not bear children incurs social disapproval [50]. So, this was indicated to have a major influence on the intentions and behaviors of women regarding spacing and contraceptive use. Hence, it is noted that effort should be put into overcoming and changing the community attitude/norms regarding FP with the focus of promoting FP use may encourage women to utilize contraceptives.

This study indicates that higher women and girls' empowerment in family planning was positively associated with modern contraceptive utilization. It was also indicated from the community-level variables that women who reside in a cluster with lower women's and girls' empowerment about family planning had lower odds of contraceptive utilization. These findings align with a study in Ethiopia revealing women's empowerment as an essential determinant of contraceptive use. Moreover, empowered women were more likely to utilize contraceptives than un-empowered women [51]. Similarly, another study in Tanzania indicated that self-efficacy for contraception was associated with utilizing modern contraceptives [45]. Women who had complete control over their healthcare were more likely to use contraceptives than women who had no control over their healthcare [44]. A systematic review also revealed that women's empowerment domains are consistently positively associated with ever use of contraceptives. In many Sub-Saharan African countries, this could also be justified as women's roles and reproductive choices could be constrained by various cultural traditions that restrict women's autonomy and encourage men's dominance [52]. This disempowering of women and girls might directly impact women's decision to utilize contraceptives.

Consistent with previous studies, this study also revealed a positive association between a community literacy level and modern contraceptive utilization. Cluster with higher women's literacy status at the community level was positively associated with using modern contraceptives. Pieces of evidence from many individual studies also indicated that women from the community with higher women's literacy status were more likely to be users of contraceptives than their reference group. Generally, women tend to have higher decision-making power regarding contraceptives by attaining higher literacy levels [53–57]. So, it is justified that being illiterate might make women unaware of the benefit of the optimal birth interval and could not overcome the misconceptions about contraceptive utilization. This might make clusters with low community literacy levels consider giving birth with a narrow birth interval as normal. The effect of community literacy in disseminating contraceptive information and counseling women is profound. The cumulative effect brings or predicts the actual use of contraceptives. This implies that a particular effort is needed to increase the number of women who are literate and, thereby, intend to utilize contraceptives.

Our study found clusters of being agrarian as a community way of life positively associated with modern contraceptive utilization. In line with these findings, pastoralist communities in

Ethiopia showed lower contraceptive utilization compared to agrarian communities [58, 59]. One factor that could contribute to the decreased use of contraceptives within pastoralist communities is the detrimental impact of husbands, religious leaders, and clan leaders on family planning decisions. Furthermore, the lack of support for family planning methods within the pastoralist community, coupled with women's limited access to education and low levels of community literacy, may also lead to lower utilization of contraceptives among this group [55, 60]. The disparity could also be the presence of improved infrastructure, schools, healthcare services, media exposure, and greater male participation in family planning decision-making among agrarian communities compared to pastoralist communities. As a result, agrarian communities may provide a more supportive setting for increased contraceptive use. Therefore, when implementing family planning interventions, it is crucial to consider each community's specific lifestyle and characteristics to ensure the program's effectiveness and relevance.

Our study also summarizes measures of the effect of cluster variables (interval IOR, POOR) and components of heterogeneity (VPC or ICC and MOR). In this case, It indicated that the clustering effect for modern contraceptive use was higher than that of individual factors. Hence, it is important to consider the community/cluster to promote the utilization of modern contraceptives. Besides, addressing and considering individual and community factors would enhance modern contraceptive utilization. Ultimately, our study results will be generalized to all reproductive-age women in Ethiopia.

## Strengths and limitations

The study has strengths and limitations. One strength is its focus on a large sample size that included unmarried women. Additionally, it utilized a summary measure of the effect of cluster variables and measures of heterogeneity components in addition to the typical interpretation of multilevel analysis. However, the study's reliance on reported contraceptive utilization may be subject to recall bias. Furthermore, not all potential factors influencing contraceptive utilization were explored due to the data limitations of a national study that aimed to cover a wide range of family planning questions.

## Conclusion

The research revealed that modern contraceptive use among women of reproductive age in Ethiopia was lower than the country's targeted contraceptive prevalence rates. Both individual and community factors played a significant role in understanding the factors associated with modern contraceptive utilization.

The study identified never being married, higher levels of education among women, and discussions about family planning at health facilities as individual-level predictors of modern contraceptive utilization. Additionally, clusters with greater empowerment of women and girls, supportive attitudes toward family planning, higher community literacy levels, and an agrarian way of life were identified as community-level predictors of modern contraceptive utilization.

## Supporting information

**S1 Checklist. STROBE statement—checklist of items that should be included in reports of observational studies.**
(DOCX)

## Acknowledgments

We have received the data from the Performance Monitoring for Action (PMA-Ethiopia), and we are grateful to PMA for providing us with this valuable data.

## Author Contributions

**Conceptualization:** Hailay Gebrekidan, Mussie Alemayehu, Gurmesa Tura Debelew.

**Data curation:** Hailay Gebrekidan, Gurmesa Tura Debelew.

**Formal analysis:** Hailay Gebrekidan, Mussie Alemayehu.

**Funding acquisition:** Hailay Gebrekidan.

**Investigation:** Hailay Gebrekidan, Gurmesa Tura Debelew.

**Methodology:** Hailay Gebrekidan, Mussie Alemayehu, Gurmesa Tura Debelew.

**Project administration:** Hailay Gebrekidan.

**Resources:** Hailay Gebrekidan.

**Software:** Hailay Gebrekidan.

**Supervision:** Hailay Gebrekidan, Mussie Alemayehu, Gurmesa Tura Debelew.

**Validation:** Hailay Gebrekidan, Mussie Alemayehu, Gurmesa Tura Debelew.

**Visualization:** Hailay Gebrekidan, Mussie Alemayehu, Gurmesa Tura Debelew.

**Writing – original draft:** Hailay Gebrekidan, Mussie Alemayehu, Gurmesa Tura Debelew.

**Writing – review & editing:** Hailay Gebrekidan, Mussie Alemayehu, Gurmesa Tura Debelew.

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
