## [Decision Letter · Decision Letter 0]

17 Oct 2023

PONE-D-23-09872Individual and community level factors associated with modern contraceptive use among women in Ethiopia: Multilevel modelling analysisPLOS ONE

Dear Dr. Gebrekidan,

Thank you for submitting your manuscript to PLOS ONE. After careful consideration, we feel that it has merit but does not fully meet PLOS ONE’s publication criteria as it currently stands. Therefore, we invite you to submit a revised version of the manuscript that addresses the points raised during the review process.

We look forward to receiving your revised manuscript.

Kind regards,

Abera Mersha, MSc.

Academic Editor

PLOS ONE

Journal Requirements:

"The funders had no role in study design, data collection and analysis, decision to publish, or preparation of the manuscript"

Important: If there are ethical or legal restrictions to sharing your data publicly, please explain these restrictions in detail. Please see our guidelines for more information on what we consider unacceptable restrictions to publicly sharing data: http://journals.plos.org/plosone/s/data-availability#loc- unacceptable-data-access-restrictions. Note that it is not acceptable for the authors to be the sole named individuals responsible for ensuring data access.

Reviewers' comments:

Reviewer's Responses to Questions

**Comments to the Author**

1. Is the manuscript technically sound, and do the data support the conclusions?

Reviewer #1: Partly

Reviewer #2: Yes

2. Has the statistical analysis been performed appropriately and rigorously? 

Reviewer #1: No

Reviewer #2: Yes

3. Have the authors made all data underlying the findings in their manuscript fully available?

Reviewer #1: Yes

Reviewer #2: No

4. Is the manuscript presented in an intelligible fashion and written in standard English?

Reviewer #1: Yes

Reviewer #2: Yes

5. Review Comments to the Author

Reviewer #1: Manuscript Number: PONE-D-23-09872Title: Individual and community level factors associated with modern contraceptive use among women in Ethiopia: Multilevel modelling analysisVersion: 2 Date: 12 August 2023Major commentIf you correct the comment given to you based on your research article study accordingly, it will be considered for publication.AbstractYour abstract too shallow especially in background and conclusion part.What about other finding and factors associated with modern contraceptive methods like socio-demographic status, fertility statusYour finding is not matched with your study protocol which it is minimized Your conclusion in the abstract is not based on your findings.BackgroundYour background is too wide you should have minimized based on journal guidelines.In paragraph 1 you have explained only married women what about unmarried but uses family planning based on your title.Your first paragraph should be defined the problem or subject matter rather than general term.Then the next paragraphs start to problem based idea and its prevalence, burden, consequence, strategies to overcome and complication from global to local. Your background not talk about general facts of your study outcome variables and to shallow, add as comment given to in your document.Your back ground amalgamated terminologies and it needs revision one by one. Most of your document lacks the references and need updating at least five year duration.If you are using secondary data, you doesn’t mention about data extraction period.Some of idea in your document it lacks clarity, typographical errors especially grammar how did you access it.Justification /rationale of the study is not clear. You have to clearly state why this study was necessary in concise and coherent way. It needs critical revision.MethodsHow you see the difference of study population and sampled population? Correct it.Focus only on the objective of the study.EDHS 2019 was mini EDHS and have you accessed all variables what you want to analyze?In Ethiopia each region had multi diversified populations, cultural, political, religious etc. how did you justified the sampling methods and stratification methods urban or rural, large region or small region. It is difficult to estimate and generalize based on your way of interpretation and sampling methods. It needs critical revision.Inclusion and exclusion criteria, dependent variable and independent variables should be clearly justified. It needs also critical revision.What are the unknown confounding factors of this study finding and think your study design?ResultHow did you categorized age category, why not based on WHO category or EDHS it self category? Is it inclusive? Because majority of rural part of community doesn’t know their real date of birth and it needs critical revision. Please carefully cross-check the numbers (percentages) in the texts and respective tables (E.g.Table 1(No education 2600 but Literacy for those with no education (2589), wealth index, total live birth etc.), please revise and put denominator in each part of table. It needs also critical revision. Is it homogenies which means cluster? What are non-cluster parts of study populations?DiscussionYour discussion part should be rewritten. Please focus only in line with your important findings and objectives of the study. Your first paragraph should be you magnitude and objective of your study then factors associated with.Factors like the marital status, education status of women is it analyzed and output of your study? How did you accessed and from where? Not found in table.Recommendation and strong conclusion should be needed.Limitation of study needed.This all are critical part of your study and you should have revise itI am not comfortable with this study finding and justification, unless you come up with strong justification, the idea is novel and what make your study stronger the only reason. Thank you!

Reviewer #2: Dear Editor,

Thanks for the opportunity to review this manuscript which is an important topic in maternal health. The data set used and the sample size is appropriate. However indicate how the women consented to participate in this study as stated. Or does the author mean women consented to participate in the PMA? Clarify.

Comments are embeded in the track changes in attached document.

Comments:

1. The problem statement that led to this study is not clear. FP is key in reducing maternal and perinatal morbidity and mortality and induced abortion related maternal deaths. This is not clearly stated.

2. The document may have been corrupted when I downloaded it. However there seems to be a cut and paste of irrelevant material on pesticides? It was abit confusing to find this within the manuscript.

3. The manuscript looked very long and can be summarised especially the discussion for the findings of the the three objectives.

4. Clearly indicate the new knowledge that this study provides on prevalence and factors influencing contraceptive use in Ethiopia

Kind regards

Lydia Mwanzia, PhD

6. PLOS authors have the option to publish the peer review history of their article (what does this mean?). If published, this will include your full peer review and any attached files.

Reviewer #1: **Yes: **Adisu Ashiko

Reviewer #2: **Yes: **Lydia Mwanzia PhD Senior Research Associate in Midwifery Liverpool School of Tropical Medicine UK

---

## [Author Response · Author response to Decision Letter 0]

13 Dec 2023

Date: Dec 1/2023

Manuscript Number: PONE-D-23-09872

Title: Individual and community level factors associated with modern contraceptive use among women in Ethiopia: Multilevel modeling analysis

By: Hailay Gebremichael Gebrekidan1,2*, Mussie Alemayehu2, Gurmesa Tura Debelew1

To: Plos one

Dear Editor,

We, the authors of this manuscript, are pleased to thank the journal editors and the reviewers for the in-depth review of the manuscript and for giving us valuable comments and suggestions that helped us improve the manuscript. The authors received no specific funding for this work. We have made a rigorous revision of the manuscript as per your questions and comments. We have included the point-by-point response in the table below, framed as reviewers' comments/questions and authors' responses. The detailed revisions and changes we made in the main document are prepared with track changes attached separately. We expect our revision will enable the manuscript to fit the journal better. In case of any additional questions, please do not hesitate to contact us anytime.

Point by point responses to Reviewers and Editors Comments 

Reviewers' comments/questions

 Authors Response

Editor 

1. Regarding financial disclosure

The grant information you provided in the “Funding information” and “Financial disclosure” sections donot match Thank you very much, Editor

The authors received no specific funding for this work and It is revised and corrected in the manuscript under the funding information section.

2. Data availability statement Thank you very much, Editor

Regarding the data availability statement, The PMA-Ethiopia data sharing policy in http//www.pmadata.org/countries/ethiopia stated that the data can only be used for the registered study and will not be shared for others. We have revised the manuscript in the data availability and ethical considerations section.

 Finally, we, the authors, are very grateful to you for taking your precious time and significantly contributing to improving the manuscript.

Reviewer #1: 

Abstract 

1. Your abstract too shallow especially in background and conclusion part. Thank you very much, reviewer

We accepted your comments and replaced them with more comprehensive words and evidence. 

2. What about other finding and factors associated with modern contraceptive methods like socio-demographic status, fertility status. Thank you again for your comment 

The individual-level factors are considered as the socio-demographic factors. 

3. Your finding is not matched with your study protocol which it is minimized Thank you very much, reviewer

It is revised and corrected as per the comments and suggestions.

4. Your conclusion in the abstract is not based on your findings Thank you for your valuable comment

It is revised as requested

Back ground 

1. Your background is too wide you should have minimized based on journal guidelines Thank you very much, reviewer

We accepted your comments and revised them as requested

2. In paragraph 1 you have explained only married women what about unmarried but uses family planning based on your title. Thank you dear reviewer for your valuable contribution

We accepted your comments and revised them as requested

3. Your first paragraph should be defined the problem or subject matter rather than general term. Thanks again, reviewer

It is revised as per the comment

4. Then the next paragraphs start to problem-based idea and its prevalence, burden, consequence, strategies to overcome and complication from global to local Thank you very much, reviewer

We accepted your comments and included other more comprehensive evidence per the comment.

5. Your background not talk about general facts of your study outcome variables and to shallow, add as comment given to in your document. Thank you dear reviewer

We accepted your comments and included other more comprehensive evidence per the comment.

6. Your back ground amalgamated terminologies and it needs revision one by one. Thanks again dear reviewer

It is revised as per the comment

7. Most of your document lacks the references and need updating at least five year duration. Thank you very much, reviewer

It is revised as per the comment, and referencing was managed accordingly

8. If you are using secondary data, you don't mention about data extraction period. Thank you very much, reviewer

We used secondary data from performance monitoring for action Ethiopia (PMA-Ethiopia) 2019 survey data. So, we are referring to the data point of the cross-sectional 2019 data. 

9. Some of idea in your document it lacks clarity, typographical errors especially grammar how did you access it. Thank you very much, reviewer

We would like to thank the reviewer for the valuable and constructive comments.

10. Justification /rationale of the study is not clear. You have to clearly state why this study was necessary in concise and coherent way Thank you very much, reviewer

We accepted your comments and included other more comprehensive evidence per the comment.

Methods: 

1. How you see the difference of study population and sampled population? Correct it. Thank you dear reviewer for your question 

All reproductive-age women were considered the study population whereas the reproductive age woman included in the sample were considered the sampled population. So, it was corrected accordingly.

2. Focus only on the objective of the study. Thank you very much, reviewer

It is corrected accordingly.

3. EDHS 2019 was mini-EDHS and have you accessed all variables what you want to analyze? Thank you again dear reviewer for your question 

We used performance monitoring for action Ethiopia (PMA-ET 2019) cross-sectional data. We do not use EDHS 2019 for analysis. We only used it as a reference material. 

4. In Ethiopia each region had multi diversified populations, cultural, political, religious etc. how did you justified the sampling methods and stratification methods urban or rural, large region or small region. It is difficult to estimate and generalize based on your way of interpretation and sampling methods. Thank you dear reviewer for your suggestion

PMA-ET was conducted in two stages; first, each region was stratified into urban and rural areas to form sampling strata. Then, samples of enumeration areas (EAs) were selected independently in each stratum in the second stage. However, the current analysis is based on the secondary data accessed from the PMA-ET 2019 data. Hence, the details about the sampling procedure are available at https://www.pmadata.org/countries/ethiopia.

5. Inclusion and exclusion criteria, dependent variable and independent variables should be clearly justified. Thank you very much for your valuable suggestion.

It is revised as per the comment

6. What are the unknown confounding factors of this study finding and think your study design? Thank you dear reviewer for your question 

During the design and sampling, PMA-Ethiopia used stratification techniques to sample enumeration areas from urban/rural strata and large and small regions to achieve the best representative sample and control the unknown confounding factors. 

This study used a multilevel logistic regression analysis to account for the unknown confounders. So, first, we organize the data in a hierarchical structure. We specify the level at which each predictor variable should be included (individual level or community level), add all variables as predictors in the model, assess the model fit and significance, and interpret the adjusted odds ratios and associated confidence intervals.

Result 

1. How did you categorized age category, why not based on WHO category or EDHS itself category? Is it inclusive? Because majority of rural part of community doesn't know their real date of birth Thank you very much, reviewer

Based on other related literature, we put the age category to easily understand for readers the advantage of age for maternal health. The age group of 15- 19 years is labeled as "young maternal age," the age group of 20-34 years is labeled as "adult maternal age," and those greater than 35 years of age are older maternal age. The experience of contraceptive practice, healthy mother, pregnancy, and babies have a direct relationship with the age of mothers. That is why we prefer to use the stated age category over the WHO/EDHS age category. 

2. Please carefully cross-check the numbers (percentages) in the texts and respective tables (E.g. Table 1(No education 2600 but Literacy for those with no education (2589), wealth index, total live birth etc.), please revise and put denominator in each part of table Thank you very much, reviewer

It is corrected accordingly, and we put a denominator in each part of the table heading.

3. Is it homogenies which means cluster? What are non-cluster parts of study populations? Thank you very much, reviewer

In PMA-ET, the word cluster is to mean Enumeration area (EA). EA is the primary sampling unit in the PMA-ET sampling period. During the 2019 PMA-ET survey, they had a total of 265 EAs, and from each EA, 35 households were randomly selected for the household and eligible female (15-49 years of age) interview. So, all the study populations were parts of a certain cluster (EA). In this study, we used the words cluster and EA interchangeably throughout the document. 

Discussion 

1. Your discussion part should be rewritten. Please focus only in line with your important findings and objectives of the study. Factors like the marital status, education status of women is it analyzed and output of your study? Thank you very much, reviewer

It is revised accordingly. Yes, both marital status and education status were analyzed and found to be significant predictors of modern contraceptive use.

2. Your first paragraph should be your magnitude and objective of your study then factors associated with. Thank you very much, reviewer

It is revised as per the comment

3. Recommendation and strong conclusion should be needed Thank you very much, reviewer

We accepted your comments, revised it, and put a strong conclusion per the comment. 

4. Limitation of the study needed Thank you very much, reviewer

It is revised as per the comment

 Finally, we, the authors, are very grateful to you for taking your precious time and significantly contributing to improving the manuscript.

Reviewer # 2 

1. The problem statement that led to this study is not clear. FP is key in reducing maternal and perinatal morbidity and mortality and induced abortion related maternal deaths. This is not clearly stated Thank you very much dear reviewer for your valuable comment

These statements have been revised, and some confusing ideas have been deleted.

2. The document may have been corrupted when I downloaded it. However there seems to be a cut and paste of irrelevant material on pesticides? It was a bit confusing to find this within the manuscript Thank you very much, reviewer

These statements have been revised, and some confusing ideas have been deleted.

3. The manuscript looked very long and can be summarized especially the discussion for the findings of the three objectives Thank you very much, reviewer

It is revised as per the comment

4. Clearly indicate the new knowledge that this study provides on prevalence and factors influencing contraceptive use in Ethiopia Thank you very much, reviewer

It is revised as per the comment

5. The data set used and the sample size is appropriate. However, indicate how the women consented to participate in this study as stated. Or does the author mean women consented to participate in the PMA? Clarify Thank you dear reviewer for your question

Consent was not required for this study since we have used publicly available PMA-Ethiopia datasets. The data were accessed from http//www.pmadata.org/countries/ethiopia after the purpose of the study was explained, and the data-sharing policy was followed. But, PMA-ET has received the consent of each respondent during the PMA-ET 2019 survey period. 

Line 15, delete the word "current" Thank you so much, dear, for your comment

We deleted the word "current" and replaced it with "The prevalence of."

Line 17, "information exchange" do you mean awareness of family planning? Thank you so much, dear, for your comment

"Information exchange" is replaced by "Discussion about family planning between women and health care providers."

Line 42, We are talking about sustainable development goals. Not MDGs. Please revise Thank you very much, reviewer

It is revised accordingly

Line 77, provide citation of these studies 

Line 108, replace the word "is" by "was" Thank you so much, dear, for your comment

We replaced the word as per your comment

Line 359, "reference number 30" sorry could be my computer that corrupted this document. But this section does not make sense Thank you very much, reviewer

It is revised per the comment and might be visible in the revised version.

Line 398, "reference number 30" This does not make sense on this manuscript. It looks corrupted Thank you very much, reviewer

The revised version might be visible

Page 22, "References" I was not able to follow the reference list. Looks corrupted for reasons I cannot understand Thank you very much, reviewer

The reference might be visible in the revised version. 

 Finally, we, the authors, are very grateful to you for taking your precious time and significantly contributing to improving the manuscript.

Sincerely, 

Hailay Gebremichael Gebrekidan (On behalf of the authors)

Jimma University, Department of reproductive and family health

Tell: +251966346756 email: hailaygeb23@gmail.com

---

## [Decision Letter · Decision Letter 1]

23 Jan 2024

PONE-D-23-09872R1Individual and community level factors associated with modern contraceptive use among women in Ethiopia: Multilevel modeling analysisPLOS ONE

Dear Dr. Gebrekidan,

Thank you for submitting your manuscript to PLOS ONE. After careful consideration, we feel that it has merit but does not fully meet PLOS ONE’s publication criteria as it currently stands. Therefore, we invite you to submit a revised version of the manuscript that addresses the points raised during the review process.

We look forward to receiving your revised manuscript.

Kind regards,

Abera Mersha, MSc.

Academic Editor

PLOS ONE

Journal Requirements:

Reviewers' comments:

Reviewer's Responses to Questions

**Comments to the Author**

1. If the authors have adequately addressed your comments raised in a previous round of review and you feel that this manuscript is now acceptable for publication, you may indicate that here to bypass the “Comments to the Author” section, enter your conflict of interest statement in the “Confidential to Editor” section, and submit your "Accept" recommendation.

Reviewer #1: All comments have been addressed

Reviewer #2: All comments have been addressed

2. Is the manuscript technically sound, and do the data support the conclusions?

Reviewer #1: Yes

Reviewer #2: Yes

3. Has the statistical analysis been performed appropriately and rigorously? 

Reviewer #1: Yes

Reviewer #2: Yes

4. Have the authors made all data underlying the findings in their manuscript fully available?

Reviewer #1: Yes

Reviewer #2: No

5. Is the manuscript presented in an intelligible fashion and written in standard English?

Reviewer #1: No

Reviewer #2: Yes

6. Review Comments to the Author

Reviewer #1: Manuscript Number: PONE-D-23-09872

Title: Individual and community level factors associated with modern contraceptive use among women in Ethiopia: Multilevel modeling analysis

Version: 3 Date: 21 January 2024

Minor comment authors

Thank you for your response. You have corrected the given comment on my side. It will be considered for publication.

Edited and accepted part of the manuscript from my comment

The abstract part included all necessary terms and critically revised

Background flow is well managed and wonderful

The methods part clarified and Justified

The result part was revised and errors were edited

The discussion part was nice and interesting

Limitation and strength parts added

Conclusion written based on findings and edited as well

As well as the edited comment you will have minimal comments that should be edited

Language part and punctuation should be considered, and a few words highlighted and red-marked.

The background part is still wide if you can do so modify it.

Reviewer #2: 1. The manuscript reads so much better after corrections have been effected.

2. The author has substantially improved on the referencing with more resent literature and citations which is commendable.

3. Conclusion on predictors of contraceptive uptake - 'The effect of clustering in modern contraceptive use was large compared to the unexplained between cluster variation and the individual-level factors' in the concluding statement does not make sense and needs to be re- written in line with study objectives.

4. consider accepting after this had been addressed.

7. PLOS authors have the option to publish the peer review history of their article (what does this mean?). If published, this will include your full peer review and any attached files.

Reviewer #1: **Yes: **Adisu Ashiko Milkamo

Reviewer #2: No

---

## [Author Response · Author response to Decision Letter 1]

8 Mar 2024

Date: March 8/2024

Manuscript Number: PONE-D-23-09872R2

Title: Individual and community level factors associated with modern contraceptive utilization among women in Ethiopia: Multilevel modeling analysis

By: Hailay Gebremichael Gebrekidan1,2*, Mussie Alemayehu2, Gurmesa Tura Debelew1

To: Plos one

Dear Editor and reviewers

We, the authors of this manuscript, are very grateful for the valuable comments and suggestions you gave us to improve the paper. Hereunder, we responded to each point raised in the second-round review, and the changes are indicated in the track change section of the revised manuscript. We have rigorously revised the manuscript based on your questions and comments. We have included the point-by-point response in the table below, framed as reviewers' comments/questions and authors' responses. The detailed revisions and changes we made in the main document are prepared with track changes attached separately. We expect our revision will enable the manuscript to fit the journal better. If you have any additional questions, please do not hesitate to contact us anytime.

Point-by-point Responses to Reviewers and Editor's Comments 

Reviewers' comments/questions

 Authors Response

Editor 

Journal requirements 

1. Please review your reference list to ensure that it is complete and correct. If you have cited papers that have been retracted, please include the rationale for doing so in the manuscript text, or remove these references and replace them with relevant current references. Any changes to the reference list should be mentioned in the rebuttal letter that accompanies your revised manuscript. If you need to cite a retracted article, indicate the article's retracted status in the References list and also include a citation and full reference for the retraction notice Thank you, very much dear Editor, for your suggestion.

We have revised the reference list per your suggestion.

2. We note that the grant information you provided in the 'Funding Information' and 'Financial Disclosure' sections do not match.

When you resubmit, please ensure that you provide the correct grant numbers for the awards you received for your study in the 'Funding Information' section. Thank you, very much dear Editor, for your question.

We did not change financial disclosure as "the authors received no specific funding for this work."

"The authors received no specific funding for this work."

A) Please clarify the sources of funding (financial or material support) for your study. List the grants or organizations that supported your study, including funding received from your institution.

B) State what role the funders took in the study. If the funders had no role in your study, please state: "The funders had no role in study design, data collection and analysis, decision to publish, or preparation of the manuscript."

C) If any authors received a salary from any of your funders, please state which authors and which funders.

D) If you did not receive any funding for this study, please state: "The authors received no specific funding for this work."

Please include your amended statements within your cover letter; we will change the online submission form on your behalf. Thank you, dear Editor, for your question.

We used a publicly available cross-sectional dataset from the Performance Monitoring for Action Ethiopia (PMA-ET) 2019 survey. 

A) No financial or material support was received from any organizations, including our own institutions. 

B) Since the authors did not receive any funding, the funding statement was updated as "The authors received no specific funding for this work. The funders had no role in study design, data collection and analysis, decision to publish, or preparation of the manuscript."

C) None of the authors of this manuscript has received any salary from the funders.

D) Since the authors did not receive any funding, we updated it as "the authors received no specific funding for this work."

4. Thank you for stating the following Funding Information in your manuscript:

"The authors received no funding for this work. Funders had no role in the design, analysis, interpretation of the results, or manuscript preparation. "

We note that you have provided funding information that is currently declared in your Funding Statement. However, funding information should not appear in any areas of your manuscript. We will only publish funding information present in the Funding Statement section of the online submission form.

"The authors received no specific funding for this work."

Please include your amended statements within your cover letter; we will change the online submission form on your behalf. Thank you for your suggestion.

We have removed the funding-related text we provided in the manuscript.

 Finally, we, the authors, are very grateful to you for taking your precious time and significantly contributing to improving the manuscript.

Reviewer #1: 

1. Language part and punctuation should be considered, and a few words highlighted and red-marked Thank you very much, reviewer

We have reviewed and edited the document for grammar and punctuation errors. 

2. The background part is still wide if you can do so modify it Thank you again for your comment 

We have revised it per the comment

Reviewer # 2 

1. Conclusion on predictors of contraceptive uptake - 'The effect of clustering in modern contraceptive use was large compared to the unexplained between cluster variation and the individual-level factors' in the concluding statement does not make sense and needs to be re- written in line with study objectives Thank you, very much dear reviewer, for your valuable comment

The statements have been revised to emphasize the study objectives, and any confusing ideas have been removed. In addition, we have incorporated the weighted demographic and prevalence results in Table 1 and the text of the main document.

 Finally, we, the authors, are very grateful to you for taking your precious time and significantly contributing to improving the manuscript.

Sincerely, 

Hailay Gebremichael Gebrekidan (On behalf of the authors)

Jimma University, Department of reproductive and family health

Tell: +251966346756 email: hailaygeb23@gmail.com

---

## [Decision Letter · Decision Letter 2]

22 Apr 2024

PONE-D-23-09872R2Individual and community level factors associated with modern contraceptive utilization among women in Ethiopia: Multilevel modeling analysisPLOS ONE

Dear Dr. Gebrekidan,

Thank you for submitting your manuscript to PLOS ONE. After careful consideration, we feel that it has merit but does not fully meet PLOS ONE’s publication criteria as it currently stands. Therefore, we invite you to submit a revised version of the manuscript that addresses the points raised during the review process.

======

We look forward to receiving your revised manuscript.

Kind regards,

Abera Mersha, MSc.

Academic Editor

PLOS ONE

Journal Requirements:

Reviewers' comments:

Reviewer's Responses to Questions

**Comments to the Author**

1. If the authors have adequately addressed your comments raised in a previous round of review and you feel that this manuscript is now acceptable for publication, you may indicate that here to bypass the “Comments to the Author” section, enter your conflict of interest statement in the “Confidential to Editor” section, and submit your "Accept" recommendation.

Reviewer #2: All comments have been addressed

2. Is the manuscript technically sound, and do the data support the conclusions?

Reviewer #2: Yes

3. Has the statistical analysis been performed appropriately and rigorously? 

Reviewer #2: Yes

4. Have the authors made all data underlying the findings in their manuscript fully available?

Reviewer #2: Yes

5. Is the manuscript presented in an intelligible fashion and written in standard English?

Reviewer #2: Yes

6. Review Comments to the Author

Reviewer #2: Minor comments:

1. Include the following abbreviations in Acronym list- HHQ, FRS, SNNP, ODK

2. Abstract indicates the use of R software version 4.1 for analysis while the text indicate use of Stata Version 16.1. Which is which? Please clarify

3.The study findings add to the knowledge on need for Contraceptive programs that target pastoralist reproductive women

4. The literature review and discussion sections have long paragraphs in pros. The authors would improve this by adding some sub-headers to enhance readability.

7. PLOS authors have the option to publish the peer review history of their article (what does this mean?). If published, this will include your full peer review and any attached files.

Reviewer #2: **Yes: **Lydia Mwanzia PhD Moi University, college of health sciences, Eldoret Kenya

---

## [Author Response · Author response to Decision Letter 2]

27 Apr 2024

Date: April 27/2024

Manuscript Number: PONE-D-23-09872R2 

Title: Individual and community level factors associated with modern contraceptive utilization among women in Ethiopia: Multilevel modeling analysis

By: Hailay Gebremichael Gebrekidan1,2*, Mussie Alemayehu2, Gurmesa Tura Debelew1

To: Plos one

Dear Editor and reviewers

We, the authors of this manuscript, are very grateful for the valuable comments and suggestions you gave us to improve the paper. Hereunder, we responded to each point raised in the third-round review, and the changes are indicated in the track change section of the revised manuscript. We have rigorously revised the manuscript based on your questions and comments. We have included the point-by-point response in the table below, framed as reviewers' comments/questions and authors' responses. The detailed revisions and changes we made in the main document are prepared with track changes attached separately. We expect our revision will enable the manuscript to fit the journal better. If you have any additional questions, please do not hesitate to contact us anytime.

Point-by-point Responses to Reviewers and Editor's Comments 

Reviewers' comments/questions

 Authors Response

Editor 

Journal requirements 

 Finally, we, the authors, are very grateful to you for taking your precious time and significantly contributing to improving the manuscript.

Reviewer #2: Lydia Mwanzia PhD Moi University, college of health sciences, Eldoret Kenya

1. Include the following abbreviations in Acronym list- HHQ, FRS, SNNP, ODK Thank you very much, reviewer

As per the comment, we have reviewed and included the remaining lists of abbreviations in the document. 

2. Abstract indicates the use of R software version 4.1 for analysis while the text indicate use of Stata Version 16.1. Which is which? Please clarify Thank you again for your comment 

We have revised it per the comment.

In Stata version 16.0, the data underwent cleaning, aggregation, and survey weighting, following which a descriptive analysis was performed utilizing the "svy" command. Subsequently, the main analysis was executed using R software version 4.1.3

3. The study findings add to the knowledge on need for Contraceptive programs that target pastoralist reproductive women Thank you, very much dear reviewer, for your valuable comment.

We have revised it per the comment

4. The literature review and discussion sections have long paragraphs in pros. The authors would improve this by adding some sub-headers to enhance readability. Thank you again for your comment 

After incorporating the feedback provided, we have revised the background and discussion sections accordingly. The paragraphs have been adjusted, and line space were added to indicate similar contents. While we have enhanced readability, we were unable to include subheadings in each paragraph due to our commitment to aligning the document with the formatting requirements outlined by the PLOS One journal guidelines.

 Finally, we, the authors, are very grateful to you for taking your precious time and significantly contributing to improving the manuscript.

Sincerely, 

Hailay Gebremichael Gebrekidan (On behalf of the authors)

Tell: +251966346756 email: hailaygeb23@gmail.com

---

## [Editor Report · Decision Letter 3]

1 May 2024

Individual and community level factors associated with modern contraceptive utilization among women in Ethiopia: Multilevel modeling analysis

PONE-D-23-09872R3

Dear Dr. Gebrekidan,

We’re pleased to inform you that your manuscript has been judged scientifically suitable for publication and will be formally accepted for publication once it meets all outstanding technical requirements.

Kind regards,

Abera Mersha, MSc.

Academic Editor

PLOS ONE